# NOT ALL TOKENS MATTER EQUALLY: UNCERTAINTY-GUIDED TEST TIME PROMPT ADAPTATION FOR VISION–LANGUAGE MODELS

## ABSTRACT

Not all prompt tokens contribute equally to generalization under distribution shift. While test-time prompt tuning provides a lightweight approach to adapt vision-language models without retraining, most methods update all prompt tokens uniformly, without considering their individual uncertainty or relevance. We introduce **SPLAT** (**S**pike and **SL**ab **P**rompt **A**daptation at **T**est time), a selective adaptation framework that adjusts the update strength of each token based on its estimated uncertainty. SPLAT uses Monte Carlo Dropout to measure token-wise epistemic uncertainty and applies a gating function to scale gradient updates accordingly. This mechanism is grounded in a probabilistic interpretation of a spike-and-slab prior, allowing each token to be softly preserved or adapted. We further derive a variational learning objective that encourages stable adaptation while preserving pretrained knowledge. Experiments on ten cross-dataset and four domain zero-shot generalization benchmarks show that SPLAT not only improves accuracy over existing test-time prompt tuning methods but also reduces unnecessary updates and provides finer-grained, token-level control during adaptation, a capability absent in prior approaches.

## 1 INTRODUCTION

Vision-language models (Jia et al., 2021; Li et al., 2022; Alayrac et al., 2022; Yu et al., 2022), notably CLIP (Radford et al., 2021), have demonstrated impressive zero-shot generalization by learning a shared embedding space across vision and language modalities. Through large-scale contrastive pretraining on image-text pairs, CLIP enables downstream tasks such as classification or retrieval by computing similarities between image features and class-specific textual prompts—eliminating the need for task-specific fine-tuning. However, this generalization largely depends on the assumption that test-time data follows a similar distribution to the training corpus. In the presence of a distribution shift, CLIP's performance degrades considerably (Minderer et al., 2021; Wortsman et al., 2022).

Test-time adaptation has emerged as a compelling solution for mitigating such performance degradation by adapting a model using unlabeled test samples during inference (Wang et al., 2021). Unlike traditional domain adaptation, test-time adaptation requires no access to source data or task supervision. Recent work has extended test-time adaptation to CLIP-style models by adapting the prompt tokens instead of the model weights, giving rise to *test-time prompt tuning* (Shu et al., 2022), a parameter-efficient paradigm that updates only the prompt embeddings while keeping the vision-language backbone frozen.

While test-time prompt tuning offers a lightweight adaptation strategy, most existing approaches (Shu et al., 2022; Xiao et al., 2025; Abdul Samadh et al., 2023) apply uniform updates to all prompt tokens—treating each token equally, regardless of its informativeness or stability. As illustrated in Figure 1, such uniform tuning (left) indiscriminately adapts all tokens with full update magnitude, potentially leading to three limitations: (1) it introduces redundant computation by updating tokens that are already stable or uninformative; (2) it risks overfitting to spurious patterns by overwriting pretrained generalization priors; and (3) it fails to accommodate sample-specific variability—different inputs may require adapting different subsets of prompt tokens. In contrast, our approach (right)

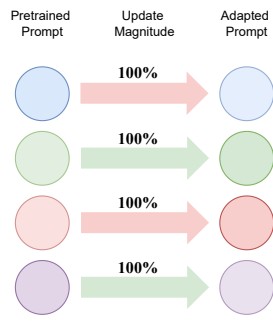 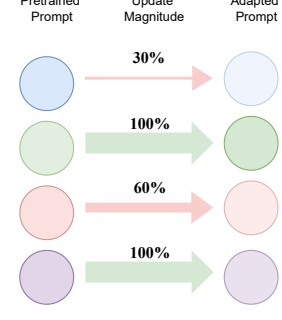

(a) Uniform Prompt Adaptation    (b) Selective Prompt Adaptation

Figure 1: **Motivation for selective prompt adaptation.** (a) Standard test-time prompt tuning methods apply uniform updates to all prompt tokens, ignoring differences in their relevance or stability. (b) In contrast, our method (SPLAT) performs token-wise selective adaptation: each token is updated with a magnitude proportional to its epistemic uncertainty. This strategy avoids unnecessary changes to stable tokens, reduces overfitting, and enables more efficient test-time adaptation.

selectively adjusts each token's update magnitude based on its test-time uncertainty, allowing stable tokens to remain unchanged while focusing adaptation on those that are truly uncertain.

We propose **SPLAT** (**S**pike-and-s**L**ab **P**rompt **A**daptation at **T**est-time), a principled test-time tuning framework for uncertainty-aware, token-selective prompt adaptation. Rather than applying uniform updates, SPLAT modulates each token's update magnitude based on its estimated epistemic uncertainty—preserving stable components while adapting only those most in need of change. SPLAT builds on three key components: First, we use Monte Carlo Dropout (Gal & Ghahramani, 2016) to estimate token-level uncertainty by measuring variance across stochastic forward passes. Second, we introduce a self-gating mechanism that maps uncertainty to a continuous weight for each token, scaling its gradient-based update. Third, we formalize this behavior using a spike-and-slab prior over prompt tokens, where each token is softly gated between being frozen (*spike*) or adapted (*slab*). This leads to a variational objective that maximizes an Evidence Lower Bound (ELBO), balancing predictive fit and adaptation regularization. We conduct on ten cross-dataset and four domain zero-shot generalization benchmarks where SPLAT consistently outperforms existing test-time prompt tuning methods while requiring fewer updates. These results underscore a central insight: *not all prompt tokens should be tuned equally*—only the uncertain ones deserve stronger updates.

## 2 RELATED WORK

**Prompt tuning for vision-language models.** Prompt tuning has emerged as an effective parameter-efficient fine-tuning paradigm for adapting large-scale vision-language models (VLMs) such as CLIP (Radford et al., 2021) to downstream tasks. CoOp (Zhou et al., 2022b) introduces learnable textual prompts in place of handcrafted templates, improving performance under few-shot settings. CoCoOp (Zhou et al., 2022a) further enhances generalization by generating input-conditional prompts via a lightweight meta-network. MaPLe (Khattak et al., 2023a) extends this by injecting prompts into both the image and text encoders, while PromptSRC (Khattak et al., 2023b) incorporates self-regularization losses to reduce overfitting. ProMetaR (Park et al., 2024) leverages meta-learning to learn task-robust prompt regularization. MMRL (Guo & Gu, 2025) proposes a joint representation space with deep encoder-injected tokens to separate general and task-specific knowledge. Adapter-based alternatives such as CLIP-Adapter (Gao et al., 2024), Tip-Adapter (Zhang et al., 2021), and MMA (Yang et al., 2024) integrate lightweight modules for few-shot adaptation. While these methods improve sample efficiency, they rely on supervised training. In contrast, our method targets *test-time* prompt tuning, adapting models without access to training data or labels.

**Test-time prompt adaptation.** Test-time adaptation (TTA) (Wang et al., 2021; Xiao & Snoek, 2024) seeks to improve robustness under distribution shift by adapting models using unlabeled target data during inference. Prompt-based TTA has emerged as a parameter-efficient strategy

for vision–language models, updating only input prompts while keeping encoder weights frozen. Early methods such as TPT (Shu et al., 2022) adapt CoOp-style prompts via entropy minimization over augmented test views. Subsequent work has enhanced this idea along several directions: DiffTPT (Feng et al., 2023) introduces diffusion-generated views to enrich augmentation diversity, while PromptAlign (Abdul Samadh et al., 2023) proposes feature-level distribution alignment via prefix tuning. To improve stability, DynaPrompt (Xiao et al., 2025) maintains a dynamic buffer of prior prompts, and AdaPrompt (Chen et al., 2022) adopts batch-wise selection to avoid overfitting to outliers. Despite architectural differences, all these prompt-based TTA methods share a common limitation: they *uniformly* update all prompt tokens, treating each token as equally uncertain or informative. This overlooks the inherent variability in token stability and fails to account for sample-specific adaptation needs. Beyond gradient-based prompt tuning, there is a parallel line of *training-free*, gradient-free CLIP-style TTA. TDA (Karmanov et al., 2024) builds a dynamic key–value cache of test features and pseudo-labels and adapts predictions via cache lookups, without updating any model or prompt parameters. TCA (Wang et al., 2025) performs token condensation on the visual branch by introducing domain anchor tokens and merging redundant tokens for CLIP and SigLIP, again without prompt learning or gradient-based optimization. These methods operate in a streaming, cache- or statistics-based setting where state is accumulated over the target stream and performance typically improves as more test samples are seen, targeting a different efficiency regime from per-sample prompt tuning. In this work we explicitly focus on the *test-time prompt tuning* regime, where a small set of prompt parameters is updated with gradients on the target domain on top of CLIP-style prompt learners. Within this setting, we introduce token-wise *selective adaptation* guided by test-time uncertainty: by estimating per-token variance via Monte Carlo Dropout, we modulate each token's update magnitude dynamically, adapting primarily the tokens that matter while keeping stable ones closer to their pretrained values.

**Applications of spike-and-slab priors.** Spike-and-slab priors have been extensively studied in Bayesian sparse modeling (Mitchell et al., 1988; Ishwaran & Rao, 2005; Nayek et al., 2021), particularly for variable selection and regularization. Mitchell et al. (1988) first introduced the spike-and-slab prior as a mixture distribution to separate important features from irrelevant ones. Ishwaran & Rao (2005) developed an MCMC-based estimation framework, enabling practical selection in high-dimensional settings. However, classical MCMC methods scale poorly with data size, motivating the use of variational inference. Titsias & Lázaro-Gredilla (2011) proposed a variational spike-and-slab model for multi-task learning, enabling structured sparsity across tasks. Ray et al. (2020) further developed scalable variational methods with theoretical guarantees for sparse logistic regression. In deep learning, spike-and-slab priors have been used for unsupervised representation learning (Goodfellow et al., 2012), structured priors (Scheipl et al., 2012), and efficient inference in high-dimensional models (Dance & Paige, 2022). While prior work has focused primarily on feature selection or global sparsity, we adopt the spike-and-slab prior at the *token level* in the context of prompt tuning. In SPLAT, we use the prior to softly gate each prompt token between preservation *spike* and adaptation *slab*, formalized within a variational inference framework.

## 3 PRELIMINARIES

**Contrastive language-image pretraining.** CLIP (Radford et al., 2021) consists of an image encoder $\mathcal{F}$ and a text encoder $\mathcal{G}$ trained jointly using a contrastive loss over image-text pairs. Given an image $\mathbf{x}$ and a textual description (or class name) $c$, a prompt template (e.g., *"A photo of a {c}"*) is filled to form the input text $\mathbf{t}_c$. The image and text are encoded as feature vectors: $\mathbf{v} = \mathcal{F}(\mathbf{x})$ and $\mathbf{e}_c = \mathcal{G}(\mathbf{t}_c)$. The probability that image $\mathbf{x}$ belongs to class $c$ is computed via a softmax over cosine similarities:

$$P(c \mid \mathbf{x}) = \frac{\exp\left(\text{sim}(\mathbf{v}, \mathbf{e}_c)/\tau\right)}{\sum_{j=1}^{C} \exp\left(\text{sim}(\mathbf{v}, \mathbf{e}_j)/\tau\right)}, \tag{1}$$

where $\text{sim}(\cdot, \cdot)$ denotes cosine similarity, $\tau$ is a learnable temperature parameter, and $C$ denotes the number of classes.

**Prompt learning for vision-language models.** To improve downstream task performance, prompt learning replaces hand-crafted text prompts with learnable embeddings. CoOp (Zhou et al., 2022b) represents the textual prompt as a sequence of learnable vectors $\{\mathbf{e}_1, \mathbf{e}_2, \cdots, \mathbf{e}_n\}$, which are optimized via backpropagation on few-shot data. To enable deeper multimodal interaction, MaPLe (Khat-

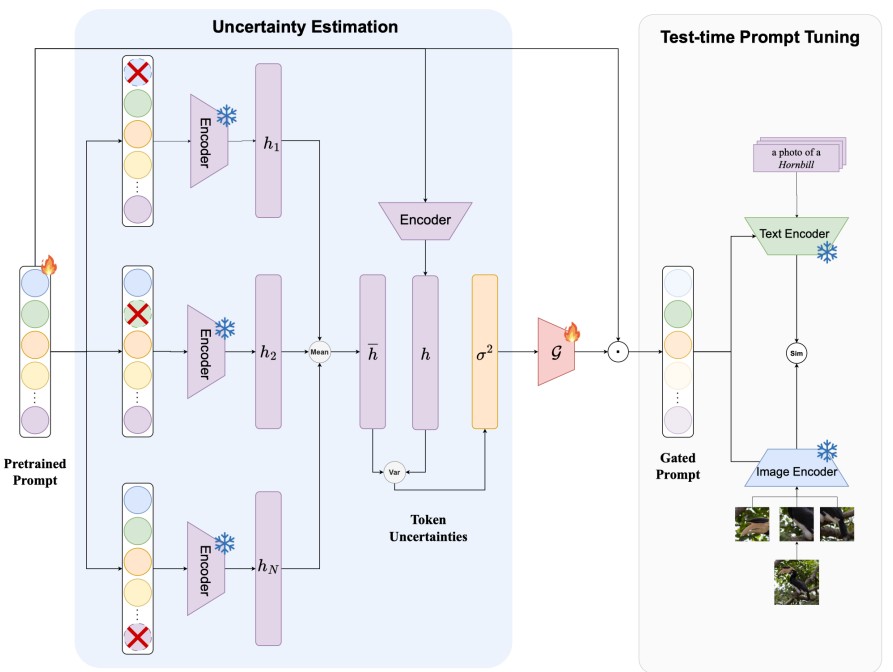

Figure 2: **Overview of SPLAT.** Given the learnable prompt tokens from a CLIP-style prompt learner, SPLAT first estimates token-wise uncertainty via Monte Carlo Dropout (left), then maps these uncertainties through a gating function to scale gradient-based updates during test-time prompt tuning (right). The *gated prompt* block represents all gated text and visual prompt tokens, while the vision and text encoders remain frozen and their embeddings are combined only through the standard similarity/classification head. The natural-language bubble is a human-readable illustration of the CLIP template and class name, rather than raw text fed directly to the encoder.

tak et al., 2023a) injects learnable prefix tokens into lower layers of both image and text encoders. However, early layers tend to encode generic features, whereas later layers capture more discriminative semantics. Building on this, MMRL (Guo & Gu, 2025) inserts modality-specific representation tokens into the higher layers of both encoders. These tokens are optimized in the training phase and used for downstream inference via similarity with class text embeddings.

**Test-time prompt tuning.** Test-time prompt tuning (TPT) (Shu et al., 2022) adapts prompt embeddings on-the-fly during inference, without labeled training data. Unlike standard prompt tuning, TPT uses unsupervised objectives—typically entropy minimization—on test samples to guide adaptation. Formally, given a test image $\mathbf{x}$ and learnable prompt embeddings $\{\mathbf{e}_1, \cdots, \mathbf{e}_n\}$, the model updates $\{\mathbf{e}_i\}$ via gradient-based optimization to minimize prediction uncertainty:

$$\min_{\{\mathbf{e}_i\}} \mathcal{L}_{\text{TPT}} = H\left(P(c \mid \mathbf{x})\right), \tag{2}$$

where $H(\cdot)$ is the entropy over softmax outputs. This approach enables adapting CLIP-like models to distribution shifts at inference time, but often treats all tokens equally, ignoring token-wise relevance or uncertainty—a problem we address in this work.

## 4 METHOD

Unlike prior TPT methods (Shu et al., 2022) that apply uniform updates to all tokens, we propose **SPLAT**, a test-time prompt tuning framework that (1) estimates the uncertainty of each prompt token via Monte Carlo (MC) Dropout, (2) uses a self-gating mechanism to modulate token-wise update strength, and (3) formalizes this process under a spike-and-slab variational inference framework. Figure 2 illustrates the full pipeline that we will detail next. SPLAT is always applied on top of an existing CLIP-style prompt learner and does not change the way inputs are fed into CLIP. For CoOp (Zhou et al., 2022b)/CoCoOp (Zhou et al., 2022a), the learnable prompts are text-side context tokens followed by the discrete class name; for MaPLe (Khattak et al., 2023a)/MMRL (Guo & Gu,

2025), the CLIP template (e.g., "a photo of a {class}") is kept fixed while virtual prompt tokens are injected into both the visual and textual branches. SPLAT operates directly on these learnable prompt embeddings (text-only or text+visual), leaving the frozen encoders and their input interface unchanged.

## 4.1 UNCERTAINTY ESTIMATION VIA MONTE CARLO DROPOUT

To selectively adapt prompt tokens at test time, we first quantify their epistemic uncertainty. Intuitively, not all tokens require equal adjustment—some are confidently aligned with the current input and domain, while others may be unstable and misaligned due to distribution shift. We aim to estimate which tokens are uncertain and thus merit stronger updates.

We adopt Monte Carlo Dropout (MC-Dropout) (Gal & Ghahramani, 2016) as a practical and architecture-agnostic Bayesian approximation to model token-level uncertainty. In our experiments, the CLIP text and vision transformers already include dropout in attention and feed-forward blocks, so SPLAT simply reuses these layers without any architectural changes. For backbones that do not expose dropout by default, one can follow standard MC-Dropout practice and insert dropout on the last few transformer blocks or on the prompt-projection layer at test time. SPLAT only requires a source of stochasticity to induce token-wise variability around the current prompt parameters; the exact location of the dropout is therefore flexible and does not require modifying the core encoder or retraining from scratch. Unlike task-level entropy or feature-level confidence, our approach focuses on prompt token embeddings themselves—capturing their stability under stochastic perturbations in the encoder.

Let $\mathbf{E} = \{\mathbf{e}_1, \ldots, \mathbf{e}_n\} \subset \mathbb{R}^d$ denote the learnable prompt embeddings, where $n$ is the number of prompt tokens and $d$ is the embedding dimension. We denote the text encoder as $\mathcal{G}(\cdot)$ and apply dropout within its layers at test time. Given a single test image-text input, we forward each prompt token $\mathbf{e}_j$ through the encoder $M$ times with dropout enabled, obtaining $M$ token-specific embeddings:

$$\mathbf{h}_j^{(i)} = \mathcal{G}_{\text{dropout}}^{(i)}(\mathbf{e}_j), \quad i = 1, \ldots, M. \tag{3}$$

We then compute the sample mean and variance of the token's embeddings:

$$\bar{\mathbf{h}}_j = \frac{1}{M} \sum_{i=1}^M \mathbf{h}_j^{(i)}, \quad \sigma_j^2 = \frac{1}{M} \sum_{i=1}^M \left\| \mathbf{h}_j^{(i)} - \bar{\mathbf{h}}_j \right\|_2^2. \tag{4}$$

Here, $\sigma_j^2$ serves as a token-wise uncertainty score for the $j$-th prompt token: a high variance indicates that the token's representation is unstable under dropout noise—i.e., it is sensitive to small changes and thus likely under-confident or misaligned with the current test domain. We use this variance as an effective token-level uncertainty signal for prompt adaptation, rather than as a pure estimate of epistemic uncertainty. In the test-time prompt-tuning setting considered here, the textual input (class name and template) is fixed and only the prompt parameters are updated, so the variability that matters most for our algorithm comes from uncertainty about these parameters rather than from input noise. Nonetheless, the MC-Dropout variance in general mixes epistemic and aleatoric components, and in the rest of the method, we always combine it with gradient information and a bounded gate, instead of relying on a strict decomposition.

Although MC-Dropout captures total predictive variance, it primarily estimates *epistemic uncertainty* in our setting. This is because the input prompt tokens are deterministic and shared across passes, meaning that input noise (a source of aleatoric uncertainty) is minimized. Consequently, the observed variance arises mainly from model uncertainty—i.e., how confidently the pretrained encoder processes each token under distribution shift. This estimation is efficient and compatible with frozen pretrained models, requiring only repeated forward passes and no gradient computation. It also operates at a finer granularity than prior test-time adaptation methods (Wang et al., 2021), enabling token-specific modulation rather than global adjustments. In the next section, we describe how these uncertainty scores guide prompt updates through a self-gating mechanism.

## 4.2 UNCERTAINTY-GATED PROMPT ADAPTATION

After estimating token-level uncertainty scores $\{\sigma_j^2\}_{j=1}^n$ via MC-Dropout, we introduce a gating mechanism to modulate the magnitude of adaptation applied to each prompt token. The key idea is

to adapt tokens proportionally to their epistemic uncertainty: unstable tokens should receive larger updates, while stable ones should be preserved to retain pretrained priors.

**Gating function.** We define a continuous, differentiable mapping from uncertainty to gating strength. For each prompt token $j$, we compute a gating coefficient $G_j \in [0, 1]$ as:

$$G_j = \sigma \left( w \cdot \log(1 + \sigma_j^2) \right), \tag{5}$$

where $\sigma(\cdot)$ is the sigmoid function, $\sigma_j^2$ is the token's uncertainty score, and $w$ is a temperature parameter (learnable or fixed). The logarithmic transformation mitigates the effect of extreme variance, while the sigmoid ensures bounded gradients and stable optimization. Higher uncertainty results in $G_j$ values closer to 1, allowing full adaptation; low-uncertainty tokens are softly frozen.

**Token-level update rule.** Let $\nabla_{\mathbf{e}_j} \mathcal{L}$ denote the gradient of the loss $\mathcal{L}$ with respect to prompt token $\mathbf{e}_j$. Rather than applying uniform updates across all tokens, we scale the gradient using $G_j$ before applying the update step:

$$\mathbf{e}_j \leftarrow \mathbf{e}_j - \eta \cdot G_j \cdot \nabla_{\mathbf{e}_j} \mathcal{L}, \tag{6}$$

where $\eta$ is the learning rate. In practice, this gating is applied at each gradient update step during test-time adaptation. This mechanism ensures that: (i) the adaptation is token-specific; (ii) tokens with high epistemic uncertainty—indicating instability across stochastic model realizations—receive larger updates; (iii) low-uncertainty tokens, which tend to encode generalizable or stable priors, are softly preserved.

**Discussion on the gating bias.** SPLAT uses token-wise uncertainty as a proxy for how much a token still needs to adapt at test time, rather than blindly pushing all high-variance tokens. Since MC-Dropout is applied to a fixed image–text pair, the predictive spread mainly reflects lack of model knowledge under the current prompt configuration, rather than input noise. The actual parameter change for token $j$ is proportional to

$$\|\Delta \mathbf{e}_j\| \ \propto \ G_j \, \|\nabla_{\mathbf{e}_j} \mathcal{L}\|, \tag{7}$$

so uncertainty and gradient magnitude jointly determine the update: even if $u_j$ is relatively large, a near-zero gradient keeps the update small, and high-uncertainty but low-influence tokens do not dominate adaptation. A small positive floor in $G_j$ further ensures that very "certain" tokens are not completely frozen, which helps correct cases where the model is confident but wrong.

Compared to hard selection or handcrafted rules, our gating mechanism thus provides a smooth, data-driven way to control token-wise updates with minimal computational overhead and can be seamlessly integrated into existing prompt-tuning workflows. While gradient magnitude reflects the local sensitivity of the loss to token perturbations for a specific input, uncertainty captures a token's expected instability across multiple stochastic model realizations, offering a more input-agnostic indicator of which tokens should be adapted. This probabilistic view of gating naturally connects to the classical *spike-and-slab prior*, where each token is softly modulated between a "frozen" (spike) and "adapted" (slab) state, a connection we make explicit in the following variational formulation.

### 4.3 SPIKE-AND-SLAB PRIOR AND VARIATIONAL OBJECTIVE

To provide a principled Bayesian foundation for our uncertainty-aware gating mechanism, we frame SPLAT's selective adaptation within a spike-and-slab prior framework. Originally developed for Bayesian variable selection (Andersen et al., 2014), spike-and-slab models promote sparsity by combining a concentrated *spike* (encouraging parameters to stay fixed) and a broader *slab* (allowing flexible adaptation).

**Posterior.** At test time, we view each prompt token embedding $\mathbf{e}_j$ as a latent variable drawn from an approximate posterior:

$$q(\mathbf{e}_j \mid \mathbf{x}) = \mathcal{N}(\mu_j, \sigma_j^2 \mathbf{I}), \tag{8}$$

where $\mu_j$ denotes the adapted prompt embedding, and $\sigma_j^2$ is the uncertainty estimated via MC-Dropout. This formulation captures our belief about how much the token should shift from its pretrained initialization, based on the input $\mathbf{x}$.

**Spike-and-slab prior.** We impose the following prior over each token:

$$p(\mathbf{e}_j) = \pi \cdot \mathcal{N}(\mathbf{e}_j \mid \mathbf{e}_j^0, \delta^2 \mathbf{I}) + (1 - \pi) \cdot \delta(\mathbf{e}_j - \mathbf{e}_j^0), \tag{9}$$

where $\mathbf{e}_j^0$ is the pretrained token embedding, $\delta^2$ is a relatively broader "slab" variance, and $\pi \in [0, 1]$ controls the probability of entering the slab (i.e., being adapted). The Dirac mass $\delta(\mathbf{e}_j - \mathbf{e}_j^0)$ represents the "spike" that keeps $\mathbf{e}_j$ tightly concentrated around its pretrained value. This prior enforces a soft inductive bias: most tokens are encouraged to remain unchanged (via the spike), while a sparse subset may flexibly adapt.

**Variational objective.** We optimize the following test-time ELBO:

$$\mathcal{L}_{\text{ELBO}} = \underbrace{-H\left[p(c \mid \mathbf{x}, \mathbf{e})\right]}_{\text{prediction consistency}} + \beta \cdot \underbrace{\sum_{j=1}^{n} \text{KL}\left(q(\mathbf{e}_j \mid \mathbf{x}) \,\|\, p(\mathbf{e}_j)\right)}_{\text{prompt regularization}}, \tag{10}$$

where the first term minimizes predictive entropy (as in prior TPT (Shu et al., 2022)), and the second term regularizes adapted prompt embeddings toward their pretrained priors. Here, $\beta$ balances adaptation versus stability.

Following (Louizos et al., 2017), we approximate the spike using a narrow Gaussian:

$$\delta(\mathbf{e}_j - \mathbf{e}_j^0) \approx \mathcal{N}(\mathbf{e}_j^0, \epsilon^2 \mathbf{I}), \quad \text{with } \epsilon^2 \ll \delta^2. \tag{11}$$

This allows us to compute the KL in closed form as a weighted sum:

$$\text{KL}\left(q \,\|\, p\right) \approx \pi \cdot \text{KL}\left(\mathcal{N}(\mu_j, \sigma_j^2 \mathbf{I}) \,\|\, \mathcal{N}(\mathbf{e}_j^0, \delta^2 \mathbf{I})\right) + (1 - \pi) \cdot \text{KL}\left(\mathcal{N}(\mu_j, \sigma_j^2 \mathbf{I}) \,\|\, \mathcal{N}(\mathbf{e}_j^0, \epsilon^2 \mathbf{I})\right). \tag{12}$$

Here, the first KL term corresponds to the slab component with variance $\delta^2$, which allows $\mu_j$ to move more freely, while the second term,

$$\text{KL}\left(\mathcal{N}(\mu_j, \sigma_j^2 \mathbf{I}) \,\|\, \mathcal{N}(\mathbf{e}_j^0, \epsilon^2 \mathbf{I})\right),$$

acts as the "spike" penalty: since $\epsilon^2$ is very small, this KL strongly increases when $\mu_j$ deviates from $\mathbf{e}_j^0$, effectively pulling adapted tokens back toward their pretrained embedding unless the likelihood provides sufficient evidence to move them.

This formulation justifies our gating behavior as a variational relaxation of discrete token selection: tokens with high uncertainty (large $\sigma_j^2$) naturally deviate from the spike and are softly encouraged toward the slab; stable tokens remain tightly clustered around their pretrained initialization. Unlike heuristic test-time updates, our method provides a probabilistic justification for selective adaptation under distribution shift. By integrating uncertainty-aware tuning with a sparsity-inducing prior, SPLAT enables calibrated and data-efficient prompt refinement at inference time.

**Reliability under shift.** Our goal is not to make individual learnable tokens human-interpretable, but to make test-time prompt tuning more reliable under distribution shift. In CLIP-style prompt learning, these tokens are exactly the parameters that align the frozen text and image encoders, and in the TPT setting they are the only ones updated at test time, so they are the natural knobs for adaptation. SPLAT updates token $j$ in proportion to an uncertainty-dependent gate $g(u_j)$ and its gradient magnitude $\|\nabla_{\mathbf{e}_j} \mathcal{L}\|$: tokens that are both uncertain and influential receive larger steps, uncertain but low-influence tokens change little, and highly "certain" tokens still move slightly due to a positive floor in $g(\cdot)$. Viewed through the spike-and-slab lens, this induces data-dependent shrinkage, keeping stable tokens close to their pretrained values while allowing a subset of uncertain, useful tokens to adapt more.

## 5 EXPERIMENTS

### 5.1 EXPERIMENTAL SETUP

**15 Datasets.** We evaluate our method under two standard test-time adaptation settings, following prior work (Shu et al., 2022). *Domain Generalization.* We assess robustness under distribution shift using four widely-used out-of-distribution (OOD) variants of ImageNet (Deng et al., 2009): ImageNetV2 (Recht et al., 2019) (resampled validation set), ImageNet-Sketch (Wang et al., 2019) (hand-drawn sketches), ImageNet-A (Hendrycks et al., 2021b) (natural adversarial examples), and ImageNet-R (Hendrycks et al., 2021a) (renditions in artistic styles). *Cross-Dataset Generalization.*

Table 1: **Cross-dataset evaluation across ten datasets.** Our SPLAT consistently improves performance over the underlying prompt learning methods under the same evaluation setting, demonstrating its effectiveness as a plug-and-play module for selective adaptation.

| | Caltech | Pets | Cars | Flowers | Food101 | Aircraft | SUN397 | DTD | EuroSAT | UCF101 | Average |
|---|---|---|---|---|---|---|---|---|---|---|---|
| CLIP (Radford et al., 2021) | 93.35 | 88.25 | 65.48 | 67.44 | 83.65 | 23.67 | 62.59 | 44.27 | 42.01 | 65.13 | 63.58 |
| TPT (Shu et al., 2022) | 94.16 | 87.79 | 66.87 | 68.98 | 84.67 | 24.78 | 65.50 | 47.75 | 42.44 | 68.04 | 65.10 |
| DynaPrompt (Xiao et al., 2025) | 94.32 | 88.28 | 67.65 | 69.95 | 85.42 | 24.33 | 66.32 | 47.96 | 42.28 | 68.72 | 65.52 |
| CoOp (Zhou et al., 2022b) | 93.70 | 89.14 | 64.51 | 68.71 | 85.30 | 18.47 | 64.15 | 41.92 | **46.39** | 66.55 | 63.88 |
| CoOp + TPT (Shu et al., 2022) | 93.15 | 89.48 | 66.77 | 68.48 | 86.48 | 20.51 | 66.06 | 43.32 | 37.73 | 68.91 | 64.09 |
| CoOp + DynaPrompt (Xiao et al., 2025) | **94.40** | 90.04 | 67.35 | 69.38 | 86.45 | 21.35 | 66.17 | **46.98** | 38.55 | 69.54 | 65.02 |
| **CoOp + SPLAT** | 93.88 | **90.45** | **67.66** | 69.43 | **86.67** | 21.75 | **66.78** | 46.75 | 40.75 | **69.94** | **65.41 ± 0.13** |
| CoCoOp (Zhou et al., 2022a) | **93.79** | 90.46 | 64.90 | **70.85** | 83.97 | 22.29 | 66.89 | 45.45 | 39.23 | 68.44 | 64.63 |
| CoCoOp + TPT (Shu et al., 2022) | 88.57 | 85.33 | 59.68 | 55.31 | 80.64 | 16.89 | 60.24 | 38.93 | 48.55 | 63.35 | 59.75 |
| **CoCoOp + SPLAT** | 92.31 | **90.82** | 65.21 | 70.11 | **84.74** | 24.32 | 67.15 | 46.01 | 50.22 | 69.17 | **66.01 ± 0.11** |
| MaPLe (Khattak et al., 2023a) | 93.53 | 90.46 | 65.57 | 72.23 | 86.20 | 24.74 | 67.01 | 46.49 | 48.06 | 68.69 | 66.30 |
| MaPLe + TPT (Shu et al., 2022) | 93.59 | 90.72 | 66.50 | 72.37 | 86.64 | 24.70 | 67.54 | 45.87 | 47.80 | 69.19 | 66.49 |
| PromptAlign (Abdul Samadh et al., 2023) | 94.01 | 90.76 | 68.50 | 72.39 | 86.65 | 24.80 | 67.54 | 47.24 | 47.86 | 69.92 | 66.92 |
| MaPLe + DynaPrompt (Xiao et al., 2025) | **95.17** | 90.95 | 68.26 | **73.25** | 86.60 | 24.36 | 68.18 | 48.75 | 47.53 | 69.85 | 67.29 |
| **MaPLe + SPLAT** | 94.75 | **90.98** | **68.75** | 73.11 | **87.15** | 26.32 | **68.45** | 48.82 | 48.11 | 70.03 | **67.65 ± 0.19** |
| MMRL (Guo & Gu, 2025) | 94.67 | 91.43 | 66.10 | 72.77 | 86.40 | 26.30 | 67.57 | 45.90 | **53.10** | 68.27 | 67.25 |
| MMRL + TPT | 94.40 | 91.31 | 66.49 | 72.89 | 86.18 | 26.23 | 67.27 | 46.41 | 47.18 | 69.04 | 66.74 |
| **MMRL + SPLAT** | **95.23** | **92.03** | **67.95** | 72.98 | **87.00** | 27.37 | **68.33** | 46.69 | 51.84 | 70.18 | **67.96 ± 0.15** |

Table 2: **Domain generalization evaluation on four ImageNet variants.** SPLAT consistently boosts the performance of multiple prompt learning backbones.

| | ImageNet-V2 | ImageNet-Sketch | ImageNet-A | ImageNet-R | Average |
|---|---|---|---|---|---|
| CLIP (Radford et al., 2021) | 60.86 | 46.09 | 47.87 | 73.98 | 57.20 |
| TPT (Shu et al., 2022) | 63.45 | 47.94 | 54.77 | 77.06 | 60.81 |
| DynaPrompt (Xiao et al., 2025) | 64.67 | 48.22 | 56.17 | 78.17 | 61.81 |
| CoOp (Zhou et al., 2022b) | 64.20 | 47.99 | 49.71 | 75.21 | 59.28 |
| CoOp + TPT (Shu et al., 2022) | 66.83 | 49.29 | 57.95 | 77.27 | 62.84 |
| **CoOp + SPLAT** | **67.32** | **50.75** | **58.11** | **78.74** | **63.73 ± 0.12** |
| CoCoOp (Zhou et al., 2022a) | 64.07 | 48.75 | 50.63 | 76.18 | 59.91 |
| CoCoOp + TPT (Shu et al., 2022) | **65.85** | 48.27 | 58.47 | 78.65 | 62.56 |
| **CoCoOp + SPLAT** | 65.81 | **49.19** | **59.27** | **79.26** | **63.38 ± 0.17** |
| MaPLe (Khattak et al., 2023a) | 64.07 | 49.15 | 50.90 | 76.98 | 60.28 |
| MaPLe + TPT (Shu et al., 2022) | 64.87 | 48.16 | 58.08 | 78.12 | 62.31 |
| **MaPLe + SPLAT** | **66.17** | **51.28** | **62.42** | **79.63** | **64.88 ± 0.13** |
| MMRL (Guo & Gu, 2025) | 64.47 | 49.17 | 51.20 | 77.53 | 60.59 |
| MMRL + TPT | 64.49 | 49.09 | 50.26 | 77.31 | 60.29 |
| **MMRL + SPLAT** | **65.74** | **50.20** | **54.63** | **78.80** | **62.34 ± 0.19** |

We further evaluate generalization across ten diverse classification datasets with distribution shift from the ImageNet training set: Caltech101 (Fei-Fei et al., 2004), OxfordPets (Parkhi et al., 2012), Oxford-Flowers (Nilsback & Zisserman, 2008), StanfordCars (Krause et al., 2013), FGVC-Aircraft (Maji et al., 2013), Food101 (Bossard et al., 2014), SUN397 (Xiao et al., 2010), DTD (Cimpoi et al., 2014), EuroSAT (Helber et al., 2019), and UCF101 (Soomro et al., 2012).

**Implementation details.** Our experiments apply SPLAT on top of four representative models: CoOp (Zhou et al., 2022b), CoCoOp (Zhou et al., 2022a), MaPLe (Khattak et al., 2023a), and MMRL (Guo & Gu, 2025), all using CLIP-ViT-B/16 (Radford et al., 2021) as the visual backbone. For CoOp and CoCoOp, we follow the original few-shot setting and use the default prompt length ($n = 16$) with learnable embeddings appended to the text input. For MaPLe and MMRL, we use their multi-layer prefix tuning structures. Specifically, in MMRL, we insert $n = 5$ learnable tokens into both the visual and textual encoders from layer $J = 6$ to layer 12. All models are initialized from their respective ImageNet-pretrained weights, trained under the 16-shot regime unless otherwise specified. At test time, we apply Monte Carlo Dropout (MC-Dropout) in the text encoder to sample $M = 10$ stochastic forward passes for each prompt token. We compute token-level variance across these samples as an estimate of epistemic uncertainty, and derive gating weights using the log-sigmoid function. We adopt a single-stage test-time adaptation (TTA) loop per sample, performing entropy minimization as the primary objective. Adaptation is performed for a fixed number of 10 iterations using the AdamW optimizer, with a learning rate of $0.001$ for domain generalization and $0.0001$ for cross-dataset evaluation. All experiments are conducted on a single NVIDIA RTX 3090 GPU. We include additional experiments and ablations in the Appendix for completeness, covering

gradient-based modulation, the effect of forward pass count, alternative regularization strategies, gating comparisons, and the robustness of uncertainty estimates under input perturbations.

## 5.2 RESULTS

**Cross-dataset evaluation.** We first evaluate SPLAT under cross-dataset generalization, where a model trained on ImageNet is directly evaluated on ten diverse classification datasets with no additional training. As shown in Table 1, SPLAT consistently improves performance when applied to existing few-shot prompting methods such as MaPLe (Khattak et al., 2023a) and MMRL (Guo & Gu, 2025). Specifically, MaPLe + SPLAT achieves an average accuracy of 67.65%, outperforming MaPLe (66.30%) and its test-time variants, including TPT and DynaPrompt. On top of MMRL, SPLAT delivers a +0.71% gain over the base model and surpasses all other methods with an average accuracy of 67.96%. These results underscore SPLAT's broad compatibility and strong transferability, enabling more effective adaptation across a range of unseen domains without retraining.

**Domain generalization.** To assess robustness under distribution shift, we conduct domain generalization experiments on four out-of-distribution variants of ImageNet. Table 2 shows that SPLAT brings consistent improvements across all domains, especially on the most challenging benchmarks such as ImageNet-A and ImageNet-Sketch. When combined with MaPLe, SPLAT improves the average accuracy from 60.28% to 64.88%, significantly outperforming strong TPT baselines. Likewise, MMRL + SPLAT achieves 62.34%, offering a 1.75% boost over MMRL. Notably, SPLAT matches or exceeds the performance of all prior test-time prompt tuning methods (e.g., TPT, PromptAlign, DynaPrompt) while introducing no extra model parameters. These improvements highlight the benefit of uncertainty-aware selective adaptation and validate SPLAT as a robust and efficient plug-in for test-time tuning.

## 5.3 ABLATION STUDY

To evaluate the contribution of SPLAT's core components, we conduct systematic ablation experiments on the domain generalization benchmark using MaPLe (Khattak et al., 2023a) as the backbone. Average accuracy over four ImageNet-derived OOD datasets is reported in Table 3.

Table 3: **Ablation study on domain generalization.**

| Method | Uncertainty | Gating | KL Reg. | Accuracy (%) |
|---|---|---|---|---|
| w/o Uncertainty | × | ✓ | ✓ | 62.14 |
| w/o Gating | ✓ | × | ✓ | 63.18 |
| w/o KL Reg. | ✓ | ✓ | × | 63.72 |
| Uniform Gating | × | constant | × | 63.05 |
| SPLAT (full) | ✓ | ✓ | ✓ | **64.88** |

**Uncertainty estimation.** In SPLAT, we estimate token-level epistemic uncertainty via Monte Carlo Dropout (MC-Dropout), performing multiple stochastic forward passes through the text encoder and computing the variance of each prompt token's embedding. To ablate this component (w/o Uncertainty), we disable dropout entirely and use deterministic token embeddings—thus setting the uncertainty scores $\sigma_j^2 = 0$ for all tokens. Performance drops from 65.07% to 62.14% (–2.93%), confirming that uncertainty provides essential guidance for identifying unstable tokens.

**Gating mechanism.** SPLAT uses a soft self-gating function to map each token's uncertainty to a continuous gating value $G_j \in [0, 1]$, which scales its update during backpropagation. In the ablated variant (w/o Gating), we remove this mechanism and directly apply the same update to all prompt tokens by setting $G_j = 1$ uniformly. Accuracy decreases to 63.18% (–1.70%), indicating that without selective scaling, overfitting may occur due to unnecessary updates to stable tokens.

**KL regularization.** We regularize each token's adapted embedding via a spike-and-slab prior, implemented as a KL divergence between the variational posterior $q(\mathbf{e}_j)$ and the prior $p(\mathbf{e}_j)$ centered at the pretrained embedding. In the ablation (w/o KL Reg.), we remove this KL term from the loss and optimize only the entropy objective. The performance drops to 63.72%, suggesting that without regularization, adaptation becomes less stable and may overwrite generalizable priors.

**Gating variants.** We also test a Uniform Gating baseline, where all tokens are assigned a constant gating value ($G_j = \alpha$), independent of uncertainty. This replaces the learnable log-sigmoid function in SPLAT with a fixed scaling. Accuracy drops to 63.05%, showing that uncertainty-aware, token-specific gating is more effective than naive uniform scaling.

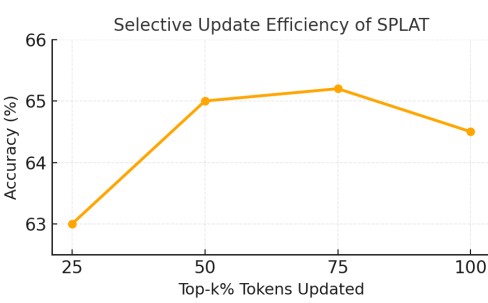

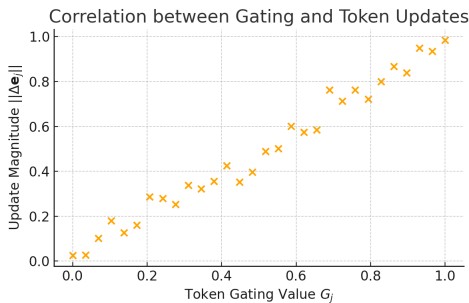

(a) **Token-wise gating visualization.**

(b) **Correlation between gating and updates.**

Figure 3: **Understanding SPLAT's token-level behavior.** (a) SPLAT adapts only high-uncertainty tokens. (b) Gating strength correlates with actual update magnitude, confirming effective modulation. Overall, these results highlight that each SPLAT component is essential: uncertainty estimation identifies which tokens require adaptation, soft gating modulates update strength proportionally, and KL regularization maintains alignment with the pretrained prior—together enabling efficient and robust test-time prompt tuning.

**Comparison with Gradient-based Modulation.** To further validate the benefit of uncertainty-guided adaptation, we compare SPLAT's gating with a gradient-based alternative, where token updates are modulated using the $\ell_2$ norm of the token gradients $|\nabla\mathbf{e}_j|$. We apply a top-k selection based on gradient magnitudes, analogous to uncertainty-based gating. While gradient magnitude is a natural signal for update strength, it can reflect batch-specific sensitivity that fluctuates across inputs. In contrast, uncertainty estimation—derived from MC-Dropout, is more focused on the token's stability under distribution shift, yielding a more global and task-aware modulation signal.

Table 4: Comparison of gating strategies on DG.

| Gating Strategy | Accuracy (%) |
|---|---|
| Gradient Norm (Top-k) | 63.55 |
| Random Gating | 62.85 |
| Inverse Gating | 63.12 |
| **Uncertainty-based (Ours)** | **64.88** |

**Qualitative Analysis.** Figure 3a illustrates how model accuracy varies when adapting only the top-$k$ tokens ranked by gating values $G_j$. Updating all tokens ($k = 100$) recovers standard TPT performance. As $k$ decreases, SPLAT selectively updates only high-uncertainty tokens, achieving peak performance around moderate $k$ (e.g., 50–75%), while updating too few tokens ($k = 25$) underfits the target domain. This trend confirms that uncertainty-based gating effectively prioritizes tokens that matter most, allowing the model to adapt with fewer updates while preserving the stability of low-uncertainty tokens.

**Gating-Update Correlation.** To verify whether our gating mechanism effectively controls adaptation strength, we measure the correlation between each token's gating value $G_j$ and the actual update magnitude $\|\Delta\mathbf{e}_j\|$ applied during optimization. As shown in Figure 3b, we observe a strong positive correlation: tokens with higher gating values receive larger gradient-based updates, while tokens with near-zero $G_j$ remain largely unchanged. This confirms that SPLAT's uncertainty-guided gating mechanism operates as intended—modulating update intensity in a continuous and differentiable manner. Such behavior is crucial for preserving pretrained knowledge in stable tokens while directing adaptation capacity toward uncertain, domain-specific ones.

## 6 CONCLUSION

We propose **SPLAT**, a test-time prompt adaptation method that updates prompt tokens selectively based on estimated uncertainty. Instead of treating all tokens equally, SPLAT uses Monte Carlo Dropout to estimate token-wise epistemic uncertainty and applies a gating function to control update strength. This mechanism is supported by a variational formulation with a spike-and-slab prior. Experiments across multiple benchmarks show that SPLAT improves generalization in zero-shot cross-dataset and domain generalization settings. The results confirm that focusing adaptation on uncertain tokens leads to more effective and efficient prompt tuning.

**Ethics Statement.** This work proposes **SPLAT**, a selective test-time prompt adaptation framework for vision-language models. We do not collect or annotate any new human data; all experiments are conducted on *publicly available* image classification datasets (e.g., ImageNet, ImageNet-R, ImageNet-Sketch) under their respective licenses. Our method focuses solely on improving model robustness under distribution shift and does not attempt to infer or exploit demographic, identity, or sensitive information. Potential misuse could include applying the method to private or restricted imagery without proper authorization; we explicitly discourage such practices and recommend adherence to data-governance policies, licensing terms, and ethical use guidelines when deploying adapted models.

**Reproducibility Statement.** All datasets and baseline methods used in this work (e.g., CoOp, CoCoOp, MaPLe, MMRL, TPT) are publicly accessible. Detailed methodology—including uncertainty estimation, gating mechanism, variational formulation, and training procedure—is provided in Sections 4 and 5. Additional ablations on gating strategies, gradient-based modulation, forward-pass count, and uncertainty robustness are reported in the Appendix to facilitate replication. All hyperparameters, evaluation protocols, and implementation settings are documented in Section 5. We will release the full source code, pretrained models, experiment scripts, and random seeds to ensure reproducibility.

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

## A LLM USAGE STATEMENT

We used a large language model (ChatGPT) solely for grammar checking and language polishing of the manuscript text. It did not contribute to research ideation, method design, experiments, data analysis, or result generation; all technical content was authored and verified by the authors.

## B ROBUSTNESS TO SYNONYM-LEVEL PERTURBATIONS

To ensure that the uncertainty measured by MC-Dropout primarily captures epistemic rather than aleatoric effects, we conduct a stability analysis under synonym substitution. Specifically, we randomly replace tokens in the prompt with semantically equivalent variants (e.g., "photo" $\rightarrow$ "picture", "a" $\rightarrow$ "one") and re-compute the token-level uncertainty scores. As shown in Figure 4, the uncertainty estimates remain stable across these perturbations, suggesting that the measured variance reflects model-driven epistemic uncertainty, not lexical noise.

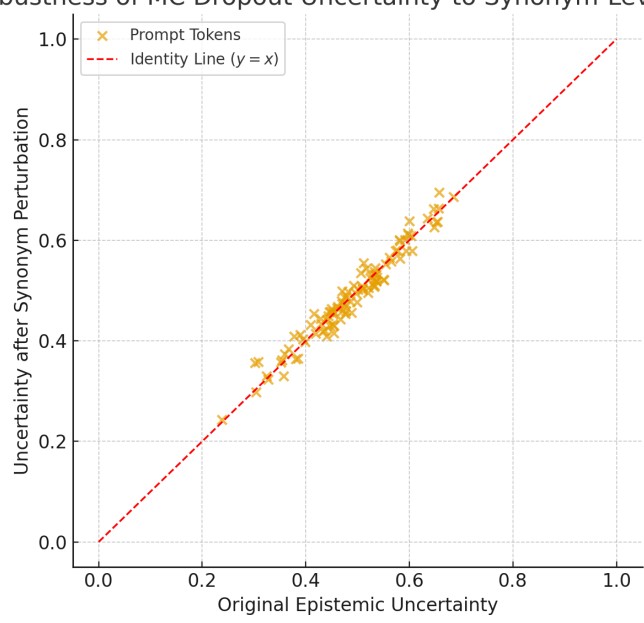

Figure 4: **Stability of MC Dropout Uncertainty.** Each dot is a prompt token, comparing its uncertainty before and after synonym substitution. Values align closely with the identity line, indicating robustness to input-level noise.

## C COMPARISON WITH ALTERNATIVE REGULARIZERS

To assess the effectiveness of our spike-and-slab regularization, we compare it against alternative regularization strategies commonly used in deep learning, such as L2 regularization and max-norm constraint. All methods are evaluated under the same MaPLe backbone with test-time prompt tuning.

Table 5: Comparison of regularization strategies on DG benchmarks.

| Regularizer | DG (avg. acc) |
| --- | --- |
| L2 Regularization | 63.75 |
| Max-Norm Constraint | 63.12 |
| No Regularization | 62.43 |
| Spike-and-Slab (Ours) | **64.88** |

As shown above, our method outperforms other regularizers by a clear margin. This result suggests that directly encouraging sparsity and selectivity via spike-and-slab prior is more aligned with the goal of adapting only the most uncertain tokens, compared to magnitude-based penalties.

## D  EFFECT OF FORWARD PASS COUNT $T$ IN MC DROPOUT

SPLAT relies on MC-Dropout only during the test-time adaptation phase to estimate token-wise uncertainty; after adaptation, inference runs deterministically with dropout disabled and has the same latency as the underlying TPT baseline. To study the trade-off between the number of stochastic forward passes $T$, accuracy, and adaptation cost, we vary $T$ in the ImageNet domain generalization setting (ImageNet $\to$ V2 / -S / -A / -R) using MaPLe+TPT with and without SPLAT. We report average accuracy over the four target domains and the adaptation time per batch on a single RTX 3090, normalized by the $T{=}1$ case.

Table 6: Effect of varying the number of MC-Dropout forward passes $T$ on domain generalization accuracy and adaptation cost.

| $T$ | DG avg. accuracy (%) | Relative adaptation cost |
|---|---|---|
| 1 | 63.35 | 1.00 |
| 2 | 64.17 | 1.52 |
| 4 | 64.28 | 2.33 |
| 8 | **64.88** | 3.31 |
| 16 | 64.73 | 7.96 |

We observe that increasing $T$ improves domain generalization accuracy but with diminishing returns. Moving from $T{=}1$ to $T{=}4$ yields a moderate gain at about $2.3\times$ adaptation cost, while $T{=}8$ gives the highest accuracy (about $+1.5$ points over $T{=}1$) at roughly $3.3\times$ cost. Further increasing to $T{=}16$ brings no additional benefit and almost doubles the cost again. In the main experiments, we therefore adopt $T{=}4$ as a practical operating point that balances robustness and adaptation overhead, while $T{=}8$ can be used when slightly better accuracy is desired and a higher adaptation cost is acceptable.

## E  ALTERNATIVE GATING STRATEGIES

We examine whether SPLAT's uncertainty-aware gating mechanism contributes meaningfully to performance by comparing it to two alternative variants: (1) random gating, where tokens are randomly selected for adaptation; and (2) inverted gating, which updates the most confident tokens instead of the least confident ones.

Table 7: Performance comparison of gating strategies.

| Gating Strategy | DG (avg. acc) |
|---|---|
| Inverted Gating | 63.12 |
| Random Gating | 62.85 |
| SPLAT (Ours) | **64.88** |

The results highlight the importance of uncertainty-based gating. In particular, the poor performance of inverted gating confirms that adapting low-uncertainty tokens is suboptimal, supporting the design intuition behind SPLAT.

## F  CORRELATION BETWEEN GATING AND GRADIENT MAGNITUDE

To address concerns about redundancy between uncertainty and gradient magnitude, we visualize the relationship between the learned gating values $G_j$ and the actual update magnitudes $\|\Delta \mathbf{e}_j\|$ across prompt tokens.

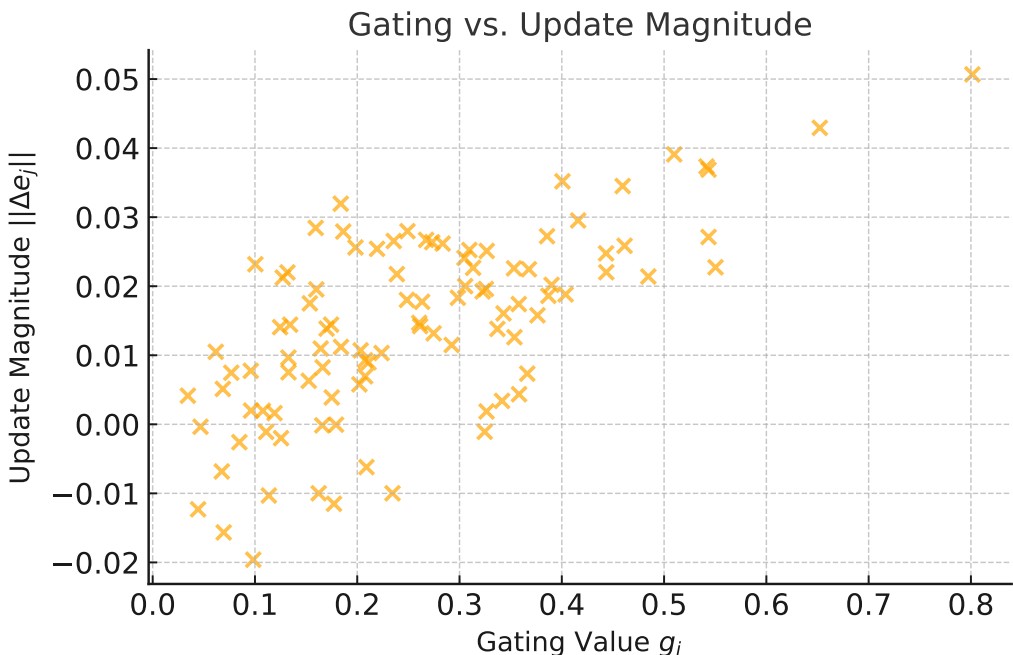

Figure 5: **Gating vs. update magnitude.** Each dot represents a prompt token. Larger $G_j$ values correlate with stronger adaptation magnitudes, confirming the intended behavior.

A clear positive trend is observed, suggesting that tokens selected by our gating mechanism are also those with larger parameter updates. This reinforces the argument that epistemic uncertainty offers a meaningful signal for selective adaptation.

## G   EFFECT OF TOKEN COUNT ON ADAPTATION

We investigate how the number of learnable prompt tokens influences SPLAT's effectiveness. The experiment controls for all other factors while varying the token count.

Table 8: Effect of number of prompt tokens.

| #Prompt Tokens | DG (avg. acc) |
| --- | --- |
| 1 | 61.91 |
| 4 | 63.12 |
| 8 | 64.05 |
| 16 | **64.88** |

Performance steadily improves as more tokens are made adaptable. However, we note that even with a small number of tokens (e.g., 4), SPLAT can still outperform vanilla TPT. This indicates that our gating strategy is effective even in low-capacity regimes.

## H   BACKBONE GENERALIZATION OF SPLAT

To check whether SPLAT extends beyond a single CLIP configuration, we evaluate MaPLe+TPT and MaPLe+SPLAT on three different frozen vision–language backbones in the ImageNet domain generalization setting (ImageNet $\rightarrow$ V2 / -S / -A / -R). We report the average accuracy over the four

Table 9: Backbone generalization of SPLAT in the domain generalization setting. SPLAT is applied on top of MaPLe+TPT and consistently improves average accuracy across all three backbones.

| Backbone | Method | Avg. acc. (%) |
|---|---|---|
| ViT-B/16 | MaPLe + TPT | 62.31 |
| ViT-B/16 | MaPLe + SPLAT | 64.88 |
| ViT-L/14 | MaPLe + TPT | 65.72 |
| ViT-L/14 | MaPLe + SPLAT | 68.08 |
| SigLIP-B | MaPLe + TPT | 65.21 |
| SigLIP-B | MaPLe + SPLAT | 67.53 |

Table 10: Comparison of different uncertainty estimators for SPLAT in the domain generalization setting (ImageNet $\rightarrow$ V2 / -S / -A / -R) on top of MaPLe+TPT. MC-Dropout achieves essentially the same accuracy as heavier alternatives, while incurring the lowest adaptation overhead and requiring no extra parameters.

| Uncertainty estimator | Extra parameters | DG avg. acc. (%) | Adapt time / batch (ms) |
|---|---|---|---|
| None (no uncertainty, TPT) | 0 | 60.29 | 18 |
| MC-Dropout ($T=4$, SPLAT) | 0 | 62.34 | 34 |
| 3-head prompt ensemble | $\approx 2\times$ params | 62.20 | 51 |
| Laplace (diagonal approximation) | 0 | 61.65 | 49 |

target domains. Across all three backbones, adding SPLAT on top of MaPLe+TPT improves average accuracy by roughly 2–2.5 points, mirroring the gains observed on CLIP ViT-B/16 alone.

# I    UNCERTAINTY ESTIMATOR ABLATION

To justify the choice of MC-Dropout in SPLAT, we ablate different uncertainty estimators in the ImageNet domain generalization setting using MaPLe+TPT as the base method. We compare four variants that differ only in how token-wise uncertainty is estimated; all other components, including the TPT objective and adaptation schedule, are kept the same. We report average accuracy over the four target domains (V2 / -S / -A / -R) and the adaptation time per batch on a single RTX 3090.

All three uncertainty-based variants improve over the "no uncertainty" baseline by about 1–2 points in average accuracy, but the gains among MC-Dropout, ensemble-style, and Laplace-style estimators are very similar. The main difference lies in cost: the ensemble requires multiple prompt heads, and the Laplace approximation needs an additional pass to estimate curvature, both of which noticeably increase adaptation time. MC-Dropout attains essentially the same accuracy with no extra parameters and the lowest overhead among the uncertainty-based methods. This supports using MC-Dropout as a practical default in SPLAT, while leaving richer estimators as an option under higher compute budgets.

