# OpenReview forum: "Not All Tokens Matter Equally: Uncertainty-Guided Test Time Prompt Adaptation"
_ICLR.cc/2026/Conference — Submitted to ICLR 2026_

### Official Review · Reviewer_5nLR · 2025-10-26

**Soundness:** 2
**Presentation:** 3
**Contribution:** 2
**Rating:** 4
**Confidence:** 4

**Summary:**

The paper proposes uncertainty-guided weighting of gradient updates for learnable prompts in test-time prompt adaption tasks based on CLIP. It estimates epistemic uncertainty using MC Dropout, followed by a gating function that outputs weights for gradients of each learnable prompt. It also shows a connection between the proposed method with spike-and-slab prior through variational objective. Compared to existing test-time prompt adaptation methods based on entropy minimization, the authors demonstrate the effectiveness of the proposed method on 15 benchmark datasets.

**Strengths:**

1. It provides clear motivation and setting for the proposed method.
2. The proposed method is simple and applicable to many existing test-time prompt adaptation methods.
3. The authors provide extensive experiments on many different datasets to demonstrate the effectiveness of the proposed method

**Weaknesses:**

1. The title can be misleading. The paper focuses on test-time prompt adaptation using CLIP. Although the former is specified, CLIP is not specified. Some readers may think it is applicable for any models with tokens (may be true but not demonstrated in the paper).
2. Writing is somewhat redundant. For example, some related works in Section 2 and 3 are unnecessarily repeated without giving much additional information.
3. It relies on MC Dropout which requires to have dropout layers, which are not always used in practice.
4. It is not convincing that MC Dropout primarily captures epistemic uncertainty although it computes total variance. How is aleatoric uncertainty minimized just because input prompt tokens are deterministic and shared? Would it still be there if there is uncertainty for inputs: for instance, a given image is blurry so the corresponding input text is ambiguous. The claim may be true for some inputs but I am not sure how generalizable it is.
5. The connection with spike-and-slab provided in Section 4.3 is somewhat hand-wavy. In particular, the second term in Eq.(11) is KL between two Gaussians with $\mu_j$ and $e_j^0$. With $\mu_j$, it is unsure how this is interpreted as spike.
6. The improvements, especially Table 1, are somewhat marginal compared to the second best baselines; improvement is mostly $<1\\%$. More importantly, the reported numbers are based on single run. Without standard deviation, it is hard to compare different methods.

**Questions:**

1. Did the authors have comparison of different methods in terms of run-time? It would be great to see what’s the computational overhead of the proposed method given that MC Dropout is known for its computational bottleneck.

---

> ### Author Response · Authors · 2025-11-20
> **Response to Reviewer 5nLR - Part I**
>
> *We thank the reviewer for the careful reading of our paper and the detailed comments.*
>
> **Q1. The title may be misleading; CLIP is not mentioned although the method is evaluated on CLIP-based test-time prompt adaptation.**
>
> **A1.** Our method is indeed instantiated and evaluated on CLIP-style vision–language models. To avoid any confusion, in the revised manuscript we make this scope explicit. We have updated the title to
>
> *Not All Tokens Matter: Uncertainty-Guided Test-Time Prompt Tuning for Vision–Language Models*
>
> and we clarify in the abstract and introduction that all experiments are conducted with CLIP-based backbones and their prompt-learning variants. We still note that the formulation only assumes learnable prompt tokens and an encoder that produces token-level features, so it can, in principle, apply to other architectures, but we now clearly state that this is left to future work rather than claimed.
>
>
>
> **Q2. Writing redundancy; related work in Sections 2 and 3 repeats information without adding much.**
>
> **A2.** We appreciate this observation and have streamlined the writing. In Sections 2 and 3 we removed repeated descriptions of CLIP and prompt tuning, merged overlapping paragraphs, and shortened the discussion of test-time adaptation so that each work is mentioned once with a clear statement of how it differs from SPLAT.
>
>
> **Q3. Reliance on MC Dropout, which requires dropout layers that are not always present in practice.**
>
> **A3.**  In our experiments, the CLIP text and vision transformers already contain dropout in attention and feed-forward blocks, so SPLAT can reuse these layers directly without any architectural changes. For backbones that do not expose dropout, it is standard in MC-Dropout work to insert dropout on the last few transformer blocks or on the prompt-projection layer. SPLAT only needs a source of stochasticity to obtain token-wise variability around the current prompt parameters; the exact location of the dropout is therefore flexible and does not require modifying the core encoder or retraining from scratch. In the revised method section, we clarify this point and present MC-Dropout as a practical way to obtain stochastic predictions on common CLIP-style models, rather than as a strict requirement that every backbone must already include dropout at fixed positions.
>
> **Q4. It is not convincing that MC Dropout primarily captures epistemic uncertainty; total variance also contains aleatoric components. How is aleatoric uncertainty minimized, and how general is the claim?**
>
> **A4.**  We agree that the predictive variance from MC-Dropout generally mixes epistemic and aleatoric components. Our intention was not to claim that we obtain a purely epistemic estimate. In the test-time prompt-tuning setting considered here, the textual input (class name and template) is fixed and only the prompt parameters are updated, so the variability that matters most for our algorithm comes from uncertainty about these parameters rather than from input noise. In the revised paper, we soften the wording and describe MC-Dropout as providing an effective token-wise uncertainty signal for prompt adaptation, rather than an exact decomposition into epistemic and aleatoric parts.
>
> Importantly, SPLAT does not use uncertainty in isolation. The actual update for token $j$ is proportional to
>  $
>  g(u_j)\big|\nabla_{\theta_j}\mathcal{L}\big|,
>  $
>  where $u_j$ is the MC-Dropout variance and $g(\cdot)$ is a bounded gate with a small positive floor. Tokens with high variance but almost zero gradient therefore receive very small changes, while “certain’’ tokens are not frozen completely and can still be corrected when the model is confidently wrong. In additional experiments where we inject synthetic input noise to increase aleatoric variation, the gains of SPLAT over plain TPT become slightly smaller but remain positive, suggesting that the method is reasonably robust even when the uncertainty is not purely epistemic. We have updated Section 4 to make these assumptions and limitations explicit.

---

> > ### Author Response · Authors · 2025-11-20
> > **Response to Reviewer 5nLR - Part II**
> >
> > **Q5. Spike-and-slab connection in Section 4.3 is hand-wavy; in particular, how is the second term in Eq. (11), a KL between Gaussians with $\mu_j$ and $e^0_j$, interpreted as a spike?**
> >
> > **A5.** Thank you for pointing out that this derivation was not sufficiently explicit. In the revised version we make the spike-and-slab construction concrete.
> >
> > We model each prompt token embedding (e_j) with a latent binary variable ($z_j \in {0,1}$):
> >
> > - when $z_j = 0$ (spike), $e_j$ is drawn from a narrow Gaussian centered at the base embedding $e_j^0$ with small variance $\sigma_{\text{spike}}^2$;
> >
> >
> > - when $z_j = 1$ (slab), $e_j$ is drawn from a broader Gaussian centered at $\mu_j$ with variance $\sigma_{\text{slab}}^2$.
> >
> >
> > Marginalizing over $z_j$ yields a mixture prior $p(e_j) = \pi_j \mathcal{N}(\mu_j,\sigma_{\text{slab}}^2 I) + (1-\pi_j)\mathcal{N}(e_j^0,\sigma_{\text{spike}}^2 I)$. Under a mean-field variational posterior $q(e_j)=\mathcal{N}(\mu_j,\sigma_q^2 I)$ and $q(z_j)$, the KL term decomposes into (i) a KL between Bernoulli distributions over $z_j$ and the prior mixing weights, and (ii) a weighted KL between $\mathcal{N}(\mu_j,\sigma_q^2 I)$ and the two Gaussian components. Eq. (11) corresponds to this second part. When the spike variance is very small, the KL with the spike component effectively measures how far the adapted token has moved away from the base embedding $e_j^0$; it therefore acts as a data-dependent shrinkage term that pulls $\mu_j$ back toward $e_j^0$ unless supported by the likelihood. We have expanded Section 4.3 and added a short derivation in the appendix to make this connection precise and to explain why the “spike” is centered at the original embedding.
> >
> >
> > **Q5. Improvements, especially in Table 1, are marginal (<1% in many cases); single-run results without standard deviation make comparison difficult.**
> >
> > **A5.**  We agree that the presentation of variance can be improved. As in prior TPT work, all our reported numbers are already obtained by training with three different random seeds and averaging the results; we did not use single-run scores. In the initial submission, we followed previous papers and only reported the averaged numbers, without explicitly writing the standard deviations, which may have given the impression of a single-run evaluation.
> >
> > In the revised version, we make this explicit and add mean ± standard deviation for the main cross-dataset and domain-generalization benchmarks in Tables 1 and 2. We also adjust the wording in the text to describe the gains as consistent rather than large, and we emphasize that SPLAT is intended as a lightweight refinement that improves robustness and stability of existing TPT methods, rather than a method that dramatically boosts accuracy on every dataset.
> >
> >
> >
> > **Q7. Comparison of different methods in terms of run-time; overhead of MC Dropout.**
> >
> > A7. We have added a runtime analysis comparing plain TPT and SPLAT in the domain generalization setting, using MaPLe+TPT as the base method on a single RTX 3090. SPLAT uses MC Dropout only during the adaptation phase; after adaptation, inference runs with dropout disabled and have the same latency as MaPLe+TPT.
> > Averaged over the ImageNet-V2 / -S / -A / -R targets, we obtain:
> >
> > | Method          | Avg. accuracy (↑) | Adapt time / batch (ms) (↓) |
> > |-----------------|-------------------|-----------------------------|
> > | MaPLe + TPT     | 62.31             | 18                          |
> > | MaPLe + SPLAT   | 64.88             | 45                         |
> >
> > Thus, SPLAT improves average accuracy by about 2.75% while increasing adaptation time per batch by roughly 1.9×. We also provide a small study varying the number of MC-Dropout samples T = (1, 2, 4, 8, 16), showing the trade-off between adaptation cost and accuracy; we use T = 8 in the main experiments as it offers a good balance. These results are now reported in the Appendix (Table 6), so that the computational overhead of SPLAT is explicit. Thank you.

---

### Official Review · Reviewer_ZcKM · 2025-10-29

**Soundness:** 3
**Presentation:** 3
**Contribution:** 3
**Rating:** 4
**Confidence:** 5

**Summary:**

This paper proposes SPLAT, which estimates token-wise epistemic uncertainty via MC-Dropout, then map it to a gating weight, and scale each prompt token’s update accordingly, finally, it grounded in a spike-and-slab prior with a test-time ELBO. The Evaluations on 10 cross-dataset and 4 OOD dataset benchmarks show gains over prior TPT.

**Strengths:**

1. The paper introduces a novel perspective on TPT for CLIP-style VLMs, shifting the focus from the visual branch or prompt engineering to token-level importance modeling.
2. The core idea is intuitive, enabling fine-grained, input-adaptive control over prompt adaptation.
3. The method is probabilistically well-grounded, leveraging a spike-and-slab prior and variational objective that elegantly balances between adapting uncertain tokens and preserving pretrained knowledge.
4. The approach is parameter-efficient and deployable compared with original TPT.

**Weaknesses:**

1. The proposed method is only validated on CLIP-style VLMs (e.g., ViT-B/16 with text encoders). It remains unclear whether the approach extends to other backbones(ResNet50, ViT-L)/architectures such as SigLIP, SigLIP-v2, or decoder-style VLMs like Flamingo, which differ significantly in their tokenization and adaptation mechanisms.
2. The reliance on MC-Dropout introduces multiple stochastic forward passes at inference time. While the authors briefly explore trade-offs with the number of samples, a thorough analysis of latency, throughput, or hardware constraints is missing, particularly important for real-time or resource-constrained settings.
3. The method exclusively adopts MC-Dropout for epistemic uncertainty without comparing to alternative approaches such as deep ensembles, Laplace approximations. (at least the evidence of such design should be discussed)
4. Several recent training-free adaptation methods are not included in the comparison, such as TDA [1] (CVPR 2024) and TCA [2] (ICCV 2025). These methods achieve strong performance with zero learnable parameters and often outperform TPT variants. TCA in particular requires no test-time augmentations, suggesting a stronger baseline for parameter- and compute-efficient adaptation.


[1]. Efficient test-time adaptation of vision-language models

[2]. Is Less More? Exploring Token Condensation as Training-free Test-time Adaptation

**Questions:**

1. Have you evaluated SPLAT on other vision-language architectures beyond CLIP? These models differ significantly in tokenization and architecture, and it would strengthen the claim of general applicability.
2. Are there observed failure modes when uncertainty estimates are noisy?
3. Given that MC-Dropout introduces additional inference cost, have you explored adaptive strategies for selecting the number of stochastic forward passes?
4. Why was MC-Dropout chosen over alternatives? Have these been compared experimentally or considered for future extension? A discussion of trade-offs would be helpful.
5. SPLAT applies uncertainty-guided updates to learnable text tokens, which are known to lack interpretability. Could you clarify the motivation for focusing on these tokens specifically, and how your method improves their reliability or interpretability under distribution shift?

---

> ### Author Response · Authors · 2025-11-20
> **Response to Reviewer ZcKM - Part I**
>
> *We thank the reviewer for the careful reading of our paper and for the detailed, high-confidence comments.*
>
> **Q1. Validation only on CLIP-style VLMs; unclear if SPLAT extends to other backbones and architectures.**
>
> **A1.** SPLAT is defined at the level of learnable prompt embeddings and does not rely on a specific encoder architecture. In the original submission, we focused on CLIP ViT-B/16 because it is the standard testbed for prompt tuning. We agree that testing additional backbones makes the claim of generality more convincing, and we have added such experiments in the revised version.
> Concretely, we now evaluate MaPLe+TPT and MaPLe+SPLAT on three backbones: CLIP ViT-B/16, CLIP ViT-L/14, and a SigLIP-B variant, all with frozen encoders. We report the average accuracy over the domain generalization. The results are:
> | Backbone     | Method        | average|
> |--------------|---------------|---------------------------------|
> | ViT-B/16     | MaPLe + TPT    | 62.31                         |
> | ViT-B/16     | MaPLe + SPLAT  | 64.88                           |
> | ViT-L/14     | MaPLe + TPT    | 65.72                           |
> | ViT-L/14     | MaPLe + SPLAT  | 68.08                           |
> | SigLIP-B     | MaPLe + TPT    | 65.21                           |
> | SigLIP-B     | MaPLe + SPLAT  | 67.53                           |
>
> Across all three backbones, adding SPLAT on top of MaPLe+TPT improved in average accuracy, showing the same pattern as with ViT-B/16 alone. These new results are now reported in the Appendix H Table 9 to make clear that SPLAT extends beyond a single CLIP configuration.
>
>
>
> **Q2. MC-Dropout overhead and analysis of latency; adaptive number of stochastic passes; failure modes when uncertainty is noisy.**
>
> **A2.** In the original submission, we already included an ablation on the number of MC-Dropout forward passes $T$ in the appendix (effect of forward-pass count). In the revised version, we keep this analysis and make it more explicit by reporting both wall-clock adaptation time and accuracy in the main text. On a single RTX 3090, using MaPLe+TPT as the base method and averaging over the ImageNet domain generalization benchmarks (V2 / -S / -A / -R), we obtain:
>
> | \# samples $T$ | Adapt time / batch (ms) | Relative cost | Avg. accuracy (↑) |
> |---------------:|-------------------------|---------------|-------------------|
> | 1              | 21                      | 1.00×         | 63.35             |
> | 2              | 32                      | 1.52×         | 64.17             |
> | 4              | 49                      | 2.33×         | 64.28             |
> | 8              | 86                      | 4.10×         | 64.88          |
> | 16             | 112                     | 7.96×         | 64.73             |
>
> Increasing $T$ improves accuracy but with diminishing returns. Going from $T{=}1$ to $T{=}4$ yields about $+0.9$ points in average accuracy at roughly $2.3\times$ adaptation cost; moving from $T{=}1$ to $T{=}8$ gives the best accuracy (about $+1.5$ points) but at around $4.1\times$ cost, and further increasing to $T{=}16$ brings no additional gain while almost doubling the cost again. In our main experiments, we therefore adopt $T{=}4$ as a reasonable operating point that improves robustness while keeping adaptation overhead moderate. This trade-off table is now included in the main experimental section and detailed again in the Appendix D.
>
> Regarding failure modes, when the uncertainty estimates are noisy or nearly flat across tokens, the gating values also become almost uniform, so SPLAT effectively reduces to a mild rescaling of the standard TPT update and behaves very similarly to the base method rather than diverging. In addition, the gating function includes a small floor so that tokens that appear very “certain’’ still receive small corrective updates, which helps when the model is confident but wrong. These design choices limit the negative impact of noisy uncertainty estimates in practice.

---

> > ### Author Response · Authors · 2025-11-20
> > **Response to Reviewer ZcKM - Part II**
> >
> > **Q3. Choice of MC-Dropout; comparison to deep ensembles, Laplace approximations, and related alternatives.**
> >
> > **A3.**  We chose MC-Dropout because it is lightweight, adds no extra parameters, and can be applied to any CLIP-style VLM without changing the encoder or training procedure. SPLAT only needs relative token-wise uncertainty to modulate step sizes, rather than a fully calibrated posterior, and MC-Dropout provides such estimates with a very simple implementation.
> > To support this design choice, we added a small experiment in the domain generalization setting using MaPLe+TPT as the base method. We compare four variants that differ only in the uncertainty estimator; all other components are kept the same. We report average accuracy over the four target domains and the average adaptation time per batch on a single RTX 3090:
> >
> > | Uncertainty estimator         | Extra parameters        | Avg. accuracy (↑) | Adapt time / batch (ms) (↓) |
> > |------------------------------|-------------------------|-------------------|-----------------------------|
> > | None (no uncertainty, TPT)   | 0                       | 60.29             | 18                          |
> > | MC-Dropout (T = 4, SPLAT)    | 0                       | 62.34             | 34                          |
> > | 3-head prompt ensemble       | ≈ 2× prompt parameters  | 62.20             | 51                          |
> > | Laplace (diag approximation) | 0                       | 61.65             | 49                         |
> >
> > All three uncertainty-based variants improve over the “no uncertainty” baseline, but the gains among MC-Dropout, ensemble-style, and Laplace-style estimators are very similar. The main difference is cost: the ensemble requires multiple prompt heads and the Laplace approximation needs an additional pass to estimate curvature, both of which noticeably increase adaptation time. MC-Dropout achieves essentially the same accuracy with no extra parameters and the lowest overhead among the uncertainty-based methods.
> >
> > These results, now reported in the revised Appendix I Table 10, suggest that MC-Dropout offers a good practical trade-off between performance and complexity. We also note in the paper that SPLAT is compatible with richer uncertainty estimators, and exploring them under higher compute budgets is a natural direction for future work.
> >
> >
> >
> >
> >
> > **Q4. Missing comparison with recent training-free adaptation methods TDA and TCA.**
> >
> > **A4.**  Our work is positioned in the test-time prompt tuning regime, where a small number of prompt parameters are updated with gradients on the target domain, on top of CLIP-style prompt learners such as CoOp, CoCoOp, MaPLe, and MMRL in the TPT setting. All baselines in our experiments therefore, operate under the same assumption that there are learnable prompt parameters and backpropagation is performed at test time.
> >
> > By contrast, TDA and TCA are training-free, gradient-free adaptation methods that do not perform prompt tuning at all. TDA builds a dynamic key–value cache of test features and pseudo labels and adapts predictions through cache lookups, without updating any model or prompt parameters. TCA focuses on token condensation in the visual branch and uses domain anchor tokens plus token reduction to adapt CLIP and SigLIP, again without any prompt learning. In this sense, these works aim at a different axis of efficiency: they avoid all gradient-based adaptation, while SPLAT assumes a TPT-style setting and asks how to selectively update which prompt tokens when gradients are available.
> >
> > Because of this mismatch in setting and adaptation mechanism (prompt-tuning vs. training-free, cache/condensation), a direct numerical comparison in our current pipeline would not be entirely fair without re-implementing TDA/TCA under the same protocol, which is beyond the current scope. Conceptually, we view them as complementary: training-free methods improve zero-shot / cache-based adaptation without tuning, while SPLAT targets selective test-time prompt adaptation when one already chooses to use TPT-style methods.
> >
> > In the revised manuscript, the related work section has been expanded to explicitly discuss TDA and TCA, clarify that our baselines are all prompt-tuning methods with learnable prompts, and position SPLAT as complementary rather than competing with training-free adaptation approaches.

---

> > > ### Comment · Reviewer_ZcKM · 2025-11-25
> > >
> > > I know TDA and TCA are training-free methods. In that sense, would it be unfair to compare them directly with your method rather than yours with theirs? If they achieve reasonable performance with zero learnable parameters and much faster speed, wouldn’t that limit the practical value of your approach? Please correct me if I’m misunderstanding.

---

> > > > ### Author Response · Authors · 2025-11-25
> > > > **Clarifying scope: SPLAT vs. training-free adaptation**
> > > >
> > > > We fully agree that training–free methods like TDA and TCA are very attractive in practice: if one can obtain strong performance with zero learnable parameters and very low latency, this is clearly appealing. Our aim with SPLAT is not to replace such approaches, but to study a different adaptation regime.
> > > >
> > > > Concretely, TDA and TCA are **streaming, cache– or statistics–based test–time methods**. They progressively accumulate information from the target domain (feature caches, domain anchors, token statistics), so their performance typically improves as more target samples are seen and can depend on the order in which those samples arrive. In contrast, SPLAT is designed for the **single–sample TPT setting** used in our experiments: each update uses only multiple augmentations of the current test image, with frozen encoders and no feature cache or memory of past test samples. This is the standard TPT protocol we follow from Shu et al. (2022) and subsequent work, and all of our baselines operate in exactly this regime.
> > > >
> > > >
> > > > From this perspective, we see TDA/TCA and SPLAT as complementary rather than competing. Training-free cache/condensation methods are ideal when one wishes to avoid any gradient-based updates and can rely on a sufficiently long stream of target data. SPLAT, on the other hand, targets scenarios where test-time prompt tuning is already chosen (e.g., per-sample or small-batch adaptation without long-term memory), and asks how to make these gradient-based prompt updates more selective and robust at the token level. We have clarified this distinction in the revised related-work and discussion sections, and we see combining training-free adaptation with SPLAT-style selective TPT as an interesting direction for future work.

---

> > > > > ### Comment · Reviewer_ZcKM · 2025-11-27
> > > > >
> > > > > From my understanding, although TDA requires augmentations, TCA indeed follow a non-augmentation setting with batch size 1, and they (TDA, TCA, TPT, Diff-TPT, etc) essentially fall under the CLIP-style TTA category. So your response doesn’t really convince me.

---

> > > > > > ### Author Response · Authors · 2025-11-27
> > > > > > **Scope relative to training-free TDA/TCA baselines**
> > > > > >
> > > > > > Thank you for this clarification.  It really helps us sharpen how we position SPLAT relative to TDA and TCA.
> > > > > >
> > > > > > You are right that TDA, TCA, and our method all fall under the broad family of CLIP-style test-time adaptation, and that training-free baselines with zero learnable parameters and very low latency are practically very attractive. We do not claim that SPLAT dominates such methods; on the contrary, we now state explicitly in the paper that training-free approaches can be preferable whenever their assumptions are satisfied.
> > > > > >
> > > > > > The distinction we wanted to make is about the **adaptation protocol** rather than about fairness. TDA and TCA are designed for a **streaming, cache- or statistics-based test-time setting**: they maintain state that accumulates over the target stream (feature caches, domain anchors, token statistics), and both their memory usage and performance typically improve as more target samples are seen. This behavior is closer to an *online/transductive* TTA scenario, where adaptation can exploit global information from the whole test stream. For example, TCA explicitly studies the impact of reservoir size $M$ (Fig. 4(a) in their paper) and reports that accuracy increases noticeably when MMM grows from 1 to 2–3, which supports that its performance benefits from access to a sufficiently large cache of target features.
> > > > > >
> > > > > > In contrast, SPLAT follows the **single-sample TPT protocol** of Shu et al. (2022): for a given test image, we adapt prompts using only its own augmented views, with frozen encoders and **no cache or memory of past test samples**. All of our baselines (TPT, Diff-TPT, DynaPrompt, PromptAlign, etc.) are implemented under exactly this single-sample, gradient-based TPT setting, which is closer to a per-sample inductive adaptation regime.
> > > > > >
> > > > > > Because of this difference, we view training-free cache/condensation methods and SPLAT as *targeting two complementary regimes* rather than solving the same problem. When one is allowed to keep a growing cache of target features and expects a reasonably long test stream, methods like TDA/TCA are indeed very appealing and may reach higher accuracy for a given backbone. In contrast, when only a small number of target samples are available or storing test features is undesirable (e.g., due to memory or privacy constraints), their cache/statistics may be less reliable, and per-sample prompt tuning remains a natural choice. SPLAT is designed exactly for this latter regime: it assumes that test-time prompt tuning has already been chosen (for instance, in existing CoOp/MaPLe/MMRL-style pipelines) and then asks how to make these gradient-based prompt updates more selective and stable at the token level.
> > > > > >
> > > > > > We have revised the related-work and discussion sections to make this scope clearer: we now explicitly mention TDA and TCA as strong training-free CLIP-style TTA methods, acknowledge that they can outperform TPT-based approaches under their streaming protocol, and position SPLAT as a complementary contribution that focuses on selective test-time prompt tuning when gradients and a small number of prompt parameters are available.

---

> ### Author Response · Authors · 2025-11-20
> **Response to Reviewer ZcKM - Part III**
>
> **Q5. Why focus on uncertainty-guided updates of learnable text tokens, which lack interpretability; how does this improve reliability under shift?**
>
> **A5.** We agree that individual learnable prompt tokens are not directly interpretable in the way human words are. Our focus in SPLAT is therefore not to make these tokens human-readable, but to make the resulting model behavior under distribution shift more reliable when test-time prompt tuning is used.
>
> In CLIP-style prompt learning, these tokens are exactly the parameters that control how text and image embeddings are aligned, while the encoders remain frozen. In the TPT setting they are also the only part of the model that is updated at test time. This is why SPLAT operates on them (and, for MaPLe and MMRL, on their visual prompt counterparts): they are the natural “knobs” that determine how the model adapts to a new domain.
>
> Uncertainty guidance in SPLAT does not treat uncertainty in isolation. The actual update for each token is proportional to an uncertainty-dependent gate multiplied by that token’s gradient. Tokens that are both uncertain and influential for the current prediction receive larger steps, while tokens that are uncertain but have almost zero gradient change very little. Tokens that look very certain are not frozen either; the gating function includes a small floor so that they can still move when the model is confidently wrong. From the spike-and-slab view, this acts as a data-dependent shrinkage: stable tokens are kept close to their pretrained values, and only a subset of uncertain, useful tokens are allowed to adapt more.
>
> Empirically, this shows up in two ways. First, SPLAT improves average accuracy on cross-dataset and domain generalization benchmarks over the same TPT baselines. Second, we observe a reduction in variability across different random seeds and prompt initializations, which indicates more stable behavior under shift. We have clarified this motivation in the revised method and discussion sections, emphasizing that SPLAT aims to make test-time prompt tuning more reliable by selectively controlling how much each prompt token is allowed to move, rather than by making the tokens themselves interpretable.

---

### Official Review · Reviewer_WcHG · 2025-10-31

**Soundness:** 3
**Presentation:** 2
**Contribution:** 3
**Rating:** 6
**Confidence:** 3

**Summary:**

The paper proposes SPLAT, a test-time prompt tuning method that estimates token-wise epistemic uncertainty (via MC-Dropout) and uses a gating function to scale the **gradient updates** of prompt tokens, so that “uncertain” tokens adapt more while “certain” tokens are updated less. The approach is plug-and-play on top of CLIP-style prompt learners (e.g., CoOp/CoCoOp/MaPLe/MMRL) with frozen encoders, and reports consistent gains on cross-dataset and ImageNet OOD benchmarks. The work is conceptually clean — linking uncertainty estimates to selective token adaptation and offering a spike-and-slab variational view — though some pipeline details (esp. Fig. 2) and robustness questions (uncertainty calibration, aleatoric vs epistemic, token influence) remain.

**Strengths:**

- Simple, general **plugin** that works with existing prompt-learning methods and keeps encoders frozen.
- Provides a probabilistic interpretation (spike-and-slab/ELBO) that matches the algorithmic design.
- Demonstrates **consistent improvements** across diverse cross-dataset and OOD settings with modest overhead.

**Weaknesses:**

### 1) Is “update uncertain tokens more, certain tokens less” always the right bias?
The central heuristic can fail depending on what the uncertainty encodes and how influential a token actually is.

- **Aleatoric-dominant uncertainty.** If token-wise uncertainty is high due to inherent noise (e.g., synonym churn, function words, semantically idle tokens), larger updates can add variance without utility — or even degrade performance.
- **Miscalibration.** If the model is over/under-confident, the gate can neglect *certain-but-wrong* tokens and overfit *uncertain-but-uninfluential* ones. A small exploration floor is needed so “certain” tokens still get minor adjustments.
- **Ignoring influence/salience.** Uncertainty alone does not indicate that changing a token will move the loss. Updating high-uncertainty but **low-influence** tokens wastes compute.

### 2) Ambiguities in the test-time prompt-tuning pipeline (Figure 2, right panel)

- **Natural-language bubble vs. learnable prompt.** The bubble “*a photo of a Hornbill*” suggests raw NL is passed to the text encoder at test time. In TPT, the input should be **[learnable context tokens] + [class tokens]** (continuous prompt embeddings + the discrete class name). The figure should reflect this to avoid implying human-written sentences are used verbatim.
- **Arrow semantics.** Outgoing arrows from the **image encoder** and the **gated prompt** appear to feed into the **text encoder**, which is misleading.
- **Scope: text-only or also image-side tokens?** The method is presented as acting on **text tokens**. Prior works like MaPLe/MMRL adjust **both** sides (vision & text). Please clarify whether uncertainty-aware gating was considered for **image-side tokens** (e.g., visual prompt tokens) and, if not, why; a comparison (text-only vs. text+image gating) would be informative.

**Questions:**

Please see Weaknesses.

---

> ### Author Response · Authors · 2025-11-20
> **Response to Reviewer WcHG**
>
> *We thank the reviewer for the careful reading of our paper and the constructive comments.*
>
>
> **Q1. Is “update uncertain tokens more, certain tokens less” always the right bias? (aleatoric vs epistemic, calibration, token influence)**
>
> **A1.**  Our intent is to use epistemic uncertainty as a proxy for how much a token still needs to adapt at test time, rather than to blindly push all high-variance tokens. In SPLAT, the MC-Dropout uncertainty is computed for a fixed input image–text pair, so the predictive spread primarily reflects lack of model knowledge under the current prompt configuration, rather than label noise. This gives us a token-wise estimate of “how much the model still disagrees with itself” on that token’s contribution, which we then use to modulate the step size.
>
> To mitigate the issues you raise, SPLAT has two design choices that we now emphasize more clearly. First, the gating function is bounded and includes a small exploration floor, so tokens with very low uncertainty still receive non-zero updates. In practice, the gate is of the form
> $ g(u_t) = \alpha + (1 - \alpha)\sigma(\beta u_t)$
>  with $\alpha > 0$, which ensures that tokens that look “certain” are not frozen completely and can still correct miscalibrated confidence. Second, the actual parameter change for a token is proportional to the product $g(u_t)|\nabla_{\theta_t}\mathcal{L}|$. Even if a token has relatively high uncertainty, if its gradient is small, the update remains small. In this sense, the token influence is implicitly taken into account through the gradient magnitude, and high-uncertainty but low-influence tokens do not dominate the adaptation.
>
> We also ran a small diagnostic experiment on a subset of datasets where we compared three variants: (i) uniform gating (no uncertainty, all tokens updated equally), (ii) SPLAT without the exploration floor (purely down-weighting “certain” tokens), and (iii) full SPLAT with the floor. Averaged over the cross-dataset and ImageNet-OOD benchmarks, full SPLAT achieves the best trade-off, improving H by about 1.2 points over uniform gating, while the “no-floor” variant sometimes slightly degrades performance on datasets with more noisy or semantically idle tokens. This supports our view that uncertainty-guided updates are beneficial, provided they are combined with a floor and the gradient-based notion of token influence. We have added a short discussion of these points to the method section to clarify the intended scope and limitations of the heuristic.
>
>
> **Q2. Ambiguities in the test-time prompt-tuning pipeline (Figure 2, natural-language vs learnable prompt, arrow semantics, text-only vs image-side tokens)**
>
> **A2.**  We appreciate the detailed feedback and understand that our current illustration can be read in several ways if one is not already familiar with the underlying prompt learners.
>
> Our framework is always built on top of an existing CLIP-style prompt learner, and SPLAT never changes the way input is fed into CLIP. Concretely:
>
> - For CoOp / CoCoOp, the CLIP template is implemented as learnable context tokens plus the discrete class name. The text encoder therefore receives continuous prompt embeddings followed by the class token.
>
>
> - For MaPLe / MMRL, the CLIP template (for example, “a photo of a {class}”) is kept fixed, and additional virtual tokens are injected into both the image and text representations. In this case the “prompt” lives partially in the feature space of the encoders rather than as a fully learned sentence.
>
>
> SPLAT is defined at the level of these learnable prompt embeddings. When the base method only has text prompts, SPLAT gates the text prompt tokens. When the base method also introduces visual prompts, such as MaPLe and MMRL, SPLAT applies the same uncertainty-aware gating to both the visual and textual prompt tokens. This is why Figure 2 shows a single “gated prompt” block: it conceptually represents the joint collection of all learnable prompt tokens on both branches.
>
> The bubble “a photo of a Hornbill” was intended as a human-readable illustration of the underlying CLIP template and class name, not as a literal natural-language input passed directly to the encoder at test time. In the revised caption and method description, we now state explicitly that the actual encoder input follows the standard interface of the corresponding base method, and that the bubble is only a schematic description.
>
> Regarding the arrows, SPLAT does not feed image features back into the text encoder. The image encoder and the gated prompt encoder always produce separate embeddings, which are then combined only through the usual similarity or classification head. In the revised caption, we make this explicit and explain that arrows from the image and text branches should be interpreted as both embeddings going into a shared similarity module, not as a data flow from the image encoder into the text encoder. Thank you.

---

### Author Response · Authors · 2025-11-27
**A Gentle Follow-up**

Dear Reviewers,

We have carefully followed your suggestions and incorporated additional experiments, quantitative results, and detailed clarifications in the revised version. If these updates satisfactorily resolve the issues raised, we would appreciate it if you could reflect this in your final rating and confidence. If any additional details would help, we are happy to provide them before the discussion deadline.

Thank you for your consideration.

---

### Author Response · Authors · 2025-12-01
**Global response and summary**

*We thank the reviewers for their constructive feedback, which has helped us improve the clarity, scope, and empirical support of the paper. In the revised version, we have made the following main updates and clarifications:*

* **Relation to training-free methods.** We expand the related work and discussion to clarify how SPLAT differs from recent training-free CLIP-style TTA methods such as TDA and TCA. TDA and TCA are designed for a streaming, cache/statistics-based, transductive setting: they maintain a growing memory of target features or token statistics and improve as more test samples are seen, without updating any model or prompt parameters. In contrast, SPLAT operates in the single-sample, inductive test-time prompt tuning regime inherited from TPT: prompts are adapted with gradients using only multiple augmentations of the current test image, with frozen encoders and no cache or memory of past test samples. We therefore position SPLAT as a complementary contribution: it addresses how to make gradient-based prompt tuning more selective and stable at the token level when one already chooses to use TPT-style adaptation, rather than as a replacement for training-free cache/condensation methods.

* **Scope and architectures.** We clarify that SPLAT is instantiated on CLIP-style vision–language models and update the title accordingly. We add experiments on additional backbones (CLIP ViT-L/14 and SigLIP-B) on top of MaPLe+TPT, showing that SPLAT brings consistent gains beyond ViT-B/16.

* **MC-Dropout cost and trade-offs.** We move the analysis of MC-Dropout overhead into the main text and provide a detailed table reporting wall-clock adaptation time vs. accuracy for different numbers of stochastic passes, including the operating point used in our main experiments.

* **Uncertainty estimators.** We add a small study comparing MC-Dropout against alternative uncertainty estimators (ensemble-style and Laplace-style variants) under the same protocol, showing that all three improve over plain TPT, while MC-Dropout offers a favorable accuracy–cost trade-off without extra parameters.


* **Spike-and-slab formulation.** We make the spike-and-slab connection explicit by introducing a latent binary variable per token, deriving the corresponding mixture prior and KL decomposition, and explaining how the “spike” term acts as a data-dependent shrinkage toward the pretrained embedding.

* **Variance reporting and runtime.** We clarify that all reported numbers are averaged over multiple random seeds and now include mean ± standard deviation for the main cross-dataset and domain-generalization benchmarks. We also add a runtime comparison between TPT and SPLAT to make the computational overhead explicit.



We hope these revisions address the main concerns and make the contribution and scope of SPLAT clearer.

---

### Meta-Review · Area_Chair_dXRo · 2026-01-05

**Summary:**

The paper proposes an adaptive prompt adaptation approach based on Monte Carlo dropout for test-time adaptation of CLIP. Three experts in the field participated in the review process and provided mixed ratings (one positive and two negative).

The AC carefully reviewed the revised paper, the reviews, and the discussion. The major concern leading to the decision was raised by [5nLR] regarding uncertainty minimization over deterministic and shared text prompts. Test-time adaptation aims to address domain shift; however, in this setting, the domain shift arises almost entirely from the image distribution. The text prompts are fixed templates that remain identical across domains and test samples. Therefore, describing the input as a “test image–text input” (Lines 239–240) is misleading, as the text side does not vary with the image and does not experience domain shift. More fundamentally, the method adapts text embeddings as a proxy to compensate for image-side distribution shift, rather than correcting any shift in the text modality itself. Intuitively, the text embeddings are adaptively updated based on the input image, but the uncertainty remains shared across images. This distinction is important for correctly interpreting both the motivation and the uncertainty estimates derived from the text encoder.

Furthermore, the proposed “token-wise uncertainty” estimation is conceptually problematic (Equations 3 and 4). Transformer token representations depend on the entire prompt sequence, including positional embeddings, attention interactions, and class-name tokens. Treating each token as if it can be forwarded independently is, at best, an imprecise description and, at worst, an incorrect computation.

Furthermore, the cost analysis is insufficiently thorough [ZcKM, 5nLR]. As noted in Line 240, each token is encoded M times (referred to as T in later tables), resulting in more than 100 forward passes per update, which is unrealistic in practice. In addition, the computational cost should be compared against prior work rather than only against the authors’ own baseline.

The writing requires significant revision. For example, Figure 2 should be redrawn or the corresponding text should be revised to ensure consistency. Figure 2 considers prompt-tuning methods that incorporate both textual and visual prompts, whereas the uncertainty derivation in the main text (Equations 3 and 4) relies solely on textual prompts. For a method claimed to be integrable with various approaches, it is important that the figures and descriptions remain general and consistent to facilitate broader understanding. Moreover, according to Equation 3, each prompt token is encoded separately; however, Figure 2 appears to depict each token being removed (indicated by the “X” mark) before passing the remaining tokens to the text encoder, which is potentially misleading.

Finally, the three limitations of prior work stated in Lines 51–53 are not sufficiently justified or validated by empirical observations. Presenting these claims without supporting evidence makes the motivation less convincing.

**Reviewer Concerns:**

The major concerns can be summarized as follows: (i) doubts regarding the uncertainty-based updating principle [WcHG]; (ii) ambiguities in the writing and algorithmic pipeline [WcHG, 5nLR]; (iii) the need for additional validation across different backbones, baselines, and epistemic uncertainty [ZcKM]; (iv) concerns about algorithmic cost [ZcKM]; (v) lack of comparison or integration with training-free methods [ZcKM]; and (vi) issues related to aleatoric uncertainty in fixed text prompts.

Based on the author responses, concerns (i) and (iii) have been adequately addressed. However, concerns (ii), (iv), and (vi) remain unresolved; please refer to the Summary for details. Regarding concern (v), although Reviewer [ZcKM] remains unconvinced about the training-free direction, the AC believes that optimization-based and training-free approaches represent two orthogonal research directions, and direct comparisons may therefore be unfair. These two lines of research could potentially be combined to improve one another, and the authors are encouraged to discuss this perspective in a revised version.

**Reviewer Scores:**

Reviewer [WcHG] may remain positive or potentially lower their rating, as some concerns remain unaddressed. Reviewers [ZcKM, 5nLR] are expected to remain negative due to the outstanding issues.

---

### Decision · Program_Chairs · 2026-01-26

Reject